# Determination of the Peptide AWRK6 in Rat Plasma by Liquid Chromatography-Tandem Mass Spectrometry (LC-MS/MS) and Its Application to Pharmacokinetics

**DOI:** 10.3390/molecules27010092

**Published:** 2021-12-24

**Authors:** Lili Jin, Haibo Ding, Volkan Degirmenci, Hongchuan Xin, Qifan Miao, Qiuyu Wang, Dianbao Zhang

**Affiliations:** 1School of Life Sciences, Liaoning University, Shenyang 110036, China; lilijin@lnu.edu.cn (L.J.); dinghaibo_jms@163.com (H.D.); westg123456@163.com (Q.M.); qiuyuwang@lnu.edu.cn (Q.W.); 2School of Engineering, University of Warwick, Coventry CV4 7AL, UK; v.degirmenci@warwick.ac.uk; 3Qingdao Institute of Bioenergy and Bioprocess Technology, Chinese Academy of Sciences, Qingdao 266101, China; xinhc@qibebt.ac.cn; 4Department of Stem Cells and Regenerative Medicine, China Medical University, Shenyang 110122, China; 5Key Laboratory of Cell Biology, National Health Commission of China, Shenyang 110122, China; 6Key Laboratory of Medical Cell Biology, Ministry of Education of China, Shenyang 110122, China

**Keywords:** peptide, AWRK6, LC-MS/MS, pharmacokinetics, rat

## Abstract

AWRK6 was a synthesized peptide developed based on the natural occurring peptide dybowskin-2CDYa, which was discovered in frog skin in our previous study. Here, a quantitative determination method for AWRK6 analysis in rat plasma by using liquid chromatography-tandem mass spectrometry (LC-MS/MS) was established and validated following U.S. FDA guidelines. A combination of plasma precipitation and liquid–liquid extraction was applied for the extraction. For pharmacokinetics study, the rats were administrated with AWRK6 via intraperitoneal and intravenous injection. The prepared plasma samples were separated on an ODS column and analyzed by tandem MS using precursor-to-product ion pairs of *m*/*z*: 533.4→84.2 for AWRK6 and *m*/*z*: 401.9→101.1 for internal standard Polymyxin B sulfate in multiple reaction monitoring mode. AWRK6 concentrations in rat plasma peaked at about 1.2 h after intraperitoneal injections at 2.35, 4.7 and 9.4 mg/kg bodyweight. The terminal half-life was around 2.8 h. The absolute bioavailability of AWRK6 was 50% after 3 doses via injection, and the apparent volume of distribution was 4.884 ± 1.736 L. The obtained determination method and pharmacokinetics profiles of AWRK6 provides a basis for further development, and forms a benchmark reference for peptide quantification.

## 1. Introduction

Peptides are considered to be promising treatment options for microbial infections and metabolic disorders [1,2]. Antimicrobial peptides have attracted much attention for their potential to combat antibiotic resistance via membrane disruption and immunomodulation [3]. For diabetes, the gut-derived glucagon-like peptide-1 (GLP-1) presented glucose-dependent plasma glucose reduction effects. However, its elimination half-life was only ~2 min which triggers the development of GLP-1 receptor agonists (GLP-1 RAs) and their long-acting analogues [4]. Developed peptides based on GLP-1 or exendin-4; e.g., dulaglutide, exenatide, liraglutide, lixisenatide, and semaglutide have been approved for diabetes and obesity treatment in multiple markets across the world [4].

Amphibian skin secretions are a rich source of bioactive peptides [1,5]. In our previous study, a series of novel peptides were discovered in frog (*Rana dybowskii*) skin, and dybowskin-2CDYa was later modified to improve the stability, acquiring a novel peptide, namely AWRK6 [6,7]. AWRK6 presented antibacterial and lipopolysaccharide-neutralizing activities, and it has been discovered that AWRK6 regulates glycolipid metabolism to improve type 2 diabetes mellitus and metabolic associated fatty liver disease (MAFLD) [8,9,10,11]. However, the pharmacokinetic profile of AWRK6 has not been fully understood and it needs to be elucidated.

In this study, a liquid chromatography-tandem mass spectrometry (LC-MS/MS) assay for AWRK6 determination was developed, and the method was applied to investigate the pharmacokinetic characteristics of AWRK6 which was administered either intraperitoneally or intravenously. This study provided an important benchmark for further modification and functionalization of AWRK6, as well as a reference for peptide detection and pharmacokinetics.

## 2. Results

### 2.1. AWRK6 Determination

By optimizing the LC parameters, an efficient and rapid separation protocol was obtained. During the LC run time of 8 min, AWRK6 and internal standard (IS, Polymyxin B sulfate, PMB) were well separated with retention time of 2.70 and 5.97 min, respectively. As presented in Figure 1, the major peaks [M+H]^4+^ (*m*/*z* = 533.4) and [M+H]^3+^ (*m*/*z* = 401.9) were observed in the Q1 full scan spectra of AWRK6 and IS separately, and the product ions of these two precursors were analyzed. The optimal *m*/*z* transition pairs 533.4/84.2 and 401.9/101.1 were used in the following analysis for AWRK6 and IS, respectively.

### 2.2. Method Validation

As shown in the chromatograms of blank, spiked, and real samples, there was no endogenous interference in plasma at the retention time of AWRK6 and IS (Figure 2). The linear calibration curve for AWRK6 was obtained by analyzing 7 standards with the concentration range of 0.05–10 μg/mL in plasma, the equation for the curve was y = 0.090541x + 3.311952 (r = 0.9957, x indicates the concentration of AWRK6 (μg/mL), and y indicates the ratio of the peak area of AWRK6 to IS). The limits of detection (LOD) and limits of quantitation (LOQ) were 0.025 and 0.050 μg/mL, respectively. The lower limit of quantification (LLOQ) was determined as the concentration of 0.050 μg/mL (relative standard deviation (RSD) < 20%) with a signal-to-noise (S/N) over 10. LLOQ was equivalent to LOQ here. As shown in Table 1, the intra-day and inter-day precision values were less than 4%, and the accuracy was 87–92.6%. The recovery of AWRK6 and IS was more than 82% and the matrix effects (−2.213%–+4.772%) were within the acceptable range. The stability of AWRK6 quality control (QC) samples (Table 2) were acceptable (relative error < 12%) under the storage condition of room temperature for 6 h, 3 freeze–thaw cycles, and 4 °C for 12 h. However, the QC samples storage at −20 °C for 7 days presented significant degradation, with relative error ranging from −16.94 to −34.74%.

### 2.3. Pharmacokinetic Study

The validated method was applied to quantify the concentrations of AWRK6 in rat plasma. The mean plasma concentrations of AWRK6 in rats following single intraperitoneal injection of AWRK6 at 2.35, 4.7 and 9.4 mg/kg bodyweight were shown in Figure 3A, and the intravenous injection at 4.7 mg/kg was illustrated in Figure 3B. After intraperitoneal injection, AWRK6 concentrations in rat plasma peaked at about 1.2 h. Thereafter it gradually decreased and were as low as below the LLOQ after 10 h. After intravenous injection, AWRK6 in plasma was cleared at about 10 h. The pharmacokinetic parameters of AWRK6 were listed in Table 3. The absolute bioavailability of AWRK6 was about 50% at the 3 doses. The apparent volume of distribution (Vd) was 4.884 ± 1.736 L, indicating a good distribution of AWRK6 in tissues and the bonding to plasma proteins. The pharmacokinetic characteristics of AWRK6 could be fitted into a two-compartment model with a weight coefficient of 1. The transport rate constants of K10, K12 and K21 were 0.918, 2.547 and 1.667 L/h, respectively. These data suggested that AWRK6 was rapidly transported to the periphery compartment after its entry into the central compartment, whereas there was little transport from the peripheral compartment to the central compartment for clearance. It was speculated that AWRK6 was mainly cleared by glomerular filtration after reaching the target organ in the target tissue, as well as enzymatically in the plasma. Thus, the distribution of AWRK6 needs to be clarified further in a follow up study.

## 3. Discussion

In this study, the LC-MS/MS scheme was optimized to determine the peptide AWRK6 in rat plasma for pharmacokinetics study. Methanol and ACN were tried as mobile phases, and AWRK6 and IS presented shorter retention time and better peak shape in ACN. In order to adjust the pH of the mobile phase and improve the peak shape, the addition of 0.1% TFA to the mobile phase led to the strong matrix effect and affected the mass spectrum response. Thus, 0.1% FA with weak acidity was added to the mobile phase, and a better mass spectrum response was obtained. Further, the gradient elution was applied to shorten the single analysis time to 8 min. According to the product ion spectra and fragmentation pattern of AWRK6 and IS, the *m*/*z* transition pairs 533.4/84.2 and 401.9/101.1 were used for analysis. The method was validated following U.S. FDA guidelines, and the findings were observed to be within the acceptable error limits. An IS with similar physicochemical properties of the determined substance was desirable. The isotope-labeled analogue of AWRK6 was not available yet. Vancomycin, PMB, berberine, and an AWRK6 derivative (SKVWKHWKKFWHKAHKLH-NH2) were examined, and PMB was chosen as an IS for its superior recovery, matrix effect and chromatographic behavior. These optimization processes were informative for the detection of AWRK6 and its derivatives.

Using the validated method, the pharmacokinetic profile of AWRK6 was analyzed following intraperitoneal or intravenous injection in rats. Previously, in our study of AWRK6 alleviating LPS induced liver injury, the mice were injected intraperitoneally at a dose of 100 nmol/kg bodyweight [12]. Calculated using the body surface area method, the corresponding administered dose for rats was 4.7 mg/kg bodyweight, which was used as the medium dose for intraperitoneal injection. Additionally, 9.4 and 2.35 mg/kg body weight were used as the high and low doses, respectively. The intravenous dose was the same as the medium dose for intraperitoneal injection. After intraperitoneal injection, the plasma drug concentration of AWRK6 increased rapidly and then decreased gradually. Combined with the data from intravenous injection, the analysis revealed bioavailability of 50%, suggesting that AWRK6 entered the blood circulation after absorption from peritoneum via portal vein and celiac venous plexus without obvious first pass effect. Exenatide and liraglutide are GLP-1 receptor agonist peptides approved for clinical use [4,13]. Compared with exenatide, the absorption of AWRK6 was slower and the retention time in the body was longer [14]. Whereas the absorption and clearance of liraglutide are much slower than those of AWRK6 and exenatide [15]. In follow-up studies, the pharmacokinetic properties of AWRK6 will be explored for enhancement through further modification and chemical modification, as well as pharmaceutical strategies such as nanosizing.

In summary, a simple and sensitive LC-MS/MS method was established and validated for the determination of AWRK6 in rat plasma. The method was applied to pharmacokinetic study following a single intraperitoneal and intravenous administration of AWRK6, and the pharmacokinetic profiles of AWRK6 were characterized. This study laid a foundation for the functionality and modification of AWRK6, and also provided a reference for the pharmacokinetic analysis of peptides.

## 4. Materials and Methods

### 4.1. Reagents and Chemicals

The peptide AWRK6 was chemically synthesized using Fmoc solid-phase synthesis method and purified using HPLC for a purity of higher than 95%, followed by verification using HPLC-MS (GL Biochem, Shanghai, China) [10]. The obtained peptide was lyophilized for storage. PMB and heparin sodium were purchased from Meilunbio (Dalian, China). Methanol and acetonitrile (ACN) were purchased from Sigma-Aldrich (Shanghai, China). Formic acid (FA) was obtained from Mreda (Beijing, China) and dichloromethane was purchased from Aibi (Shanghai, China). All chemicals used were in HPLC grade. Ultrapure water was prepared using a Milli-Q water purifying system (Millipore, MA, USA).

### 4.2. Instrumentation and Analytical Conditions

The samples were analyzed on a LC-MS/MS system consisting of an Agilent 1290 Infinity II UHPLC System coupled with an Agilent 6460 Triple Quadrupole Mass Spectrometer with an electrospray ionisation source (ESI). The Agilent MassHunter Workstation Qualitative Analysis B.06.00 (Agilent Technologies, Santa Clara, CA, USA) software was used. Chromatographic separation was performed on a Supersil ODS2 column (4.6 mm × 250 mm, 5 μm, Dalian Elite Analytical Instruments, Dalian, China). The separation conditions were 30 °C by using a FA-ACN (0.1:99.9, *v*/*v*, A) and FA-water (0.1:99.9, *v*/*v*, B) solutions at a flow rate of 0.5 mL/min. The sample injection volume was 20 μL and the autosampler was kept at 4 °C. The initial eluent A-B (20:80, *v*/*v*) was changed to A-B (30:70, *v*/*v*) in 4 min and subsequently changed to A-B (90:10, *v*/*v*) in the next 3 min in a linear gradient form and maintained at this ratio for another 1 min. PMB was used as the IS. MS analysis was carried out in the multiple reaction monitoring (MRM) mode with following operation parameters: capillary voltage, 3.5 kV; gas temperature, 325 °C; nebulizer gas pressure, 35 psi; sheath gas, 11 L/min, 350 °C; fragmentor voltages, 155 V for AWRK6 and 75 V for IS; collision energy (CE), 80 eV for AWRK6 and 40 eV for IS; precursor-to-product ion transition, *m*/*z*: 533.3→84.1 for AWRK6 and *m*/*z*: 402.1→101.1 for IS.

### 4.3. Samples Preparation

The AWRK6 and IS were dissolved in water at 1.0 mg/mL for stock solution. The AWRK6 calibration standards at 0.05, 0.1, 0.5, 1, 2, 5, and 10 μg/mL were prepared by mixing 50 μL of working solution and 100 μL of blank plasma individually. QC samples of AWRK6 at 0.1, 0.5, and 5 μg/mL were prepared by the same method. The thawed plasma samples (100 μL) were mixed with 50 μL water (or QC), 50 μL IS (1 μg/mL), and 20 μL FA-water (25:75, *v*/*v*). A combination of plasma precipitation and liquid–liquid extraction was applied for the extraction of AWRK6 in samples. After adding 200 μL methanol and 100 μL dichloromethane, the mixtures were vortexed for 30 s and centrifuged at 14,000 g for 10 min. The supernatant was filtered by 0.22 μm membrane for LC-MS/MS analysis.

### 4.4. Method Validation

The method was validated for specificity, linearity, LOD, LOQ, LLOQ, precision, accuracy, recovery, matrix effect, and stability according to U.S. FDA guidelines [16]. Specificity was determined using blank plasma samples from 6 rats. Linearity was accessed using calibration standards in triplicate within 3 consecutive days. LOD and LOQ were estimated as concentrations of AWRK6 which generated S/N value of 3 or 10, and they were verified by continuous evaluating the S/N value of 3 LOD samples or 6 LOQ samples. LLOQ was determined using 6 calibration standard samples at 0.05 μg/mL. Precision and accuracy were accessed using QC samples and repeated 6 times within 3 consecutive days. Recovery was calculated by comparing the respective peak areas of the QC samples with the blank plasma samples spiked after extraction (repeated 6 times). Matrix effect was evaluated by comparing the respective peak areas of the QC samples with standard solutions (AWRK6 and IS in water, respectively) at the same concentrations (repeated 6 times). Stability was analyzed using QC samples under different storage conditions including room temperature for 6 h, 3 freeze–thaw cycles, 4 °C for 12 h, and−20 °C for 7 days.

### 4.5. Pharmacokinetic Study

SPF-grade male Sprague Dawley rats (280–320 g, 8 weeks old) were purchased from Liaoning Changsheng Biotechology (Liaoning, China). The rats were housed under a controlled environment (temperature: 25 ± 2 °C; relative humidity: 55 ± 10%; alternating natural light/dark cycles) with free access to food and water unless otherwise noted. The study was approved by the Laboratory Animal Welfare and Ethical Review Board of China Medical University (CMU2019089, 8 March 2019). According to the previous study [12], the rats was administrated with AWRK6 dissolved in saline intraperitoneally at 2.35, 4.7, and 9.4 mg/kg bodyweight and intravenously at 4.7 mg/kg after being fasted for 12 h. The intravenous doing was administered via the tail vein of the rats. Blood samples were collected from orbital venous plexus at 0.083, 0.167, 0.25, 0.5, 1, 1.5, 2, 3, 4, 6, 8, and 10 h using heparin tubes after administration, followed by centrifugation at 4000 g for 10 min at 4 °C and storage at −80 °C. The obtained data were analyzed using DAS 2.0 software (Anhui Provincial Center for Drug Clinical Evaluation). The absolute bioavailability of AWRK6 for intraperitoneal administration was calculated by the following formula:(1)$$F=\frac{AUC_{i.p.}\times D_{i.v.}}{D_{i.p.}\times AUC_{i.v.}}\times 100\%$$

## Figures and Tables

**Figure 1 molecules-27-00092-f001:**
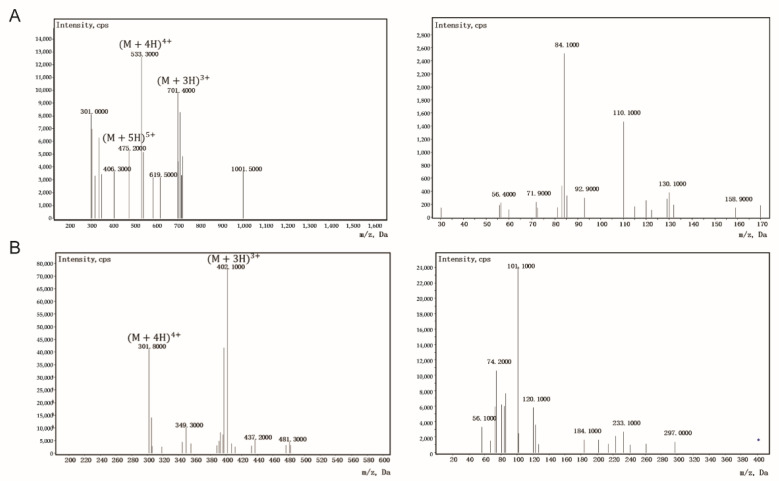
Product ion spectra and fragmentation pattern of AWRK6 (**A**) and internal standard (**B**).

**Figure 2 molecules-27-00092-f002:**
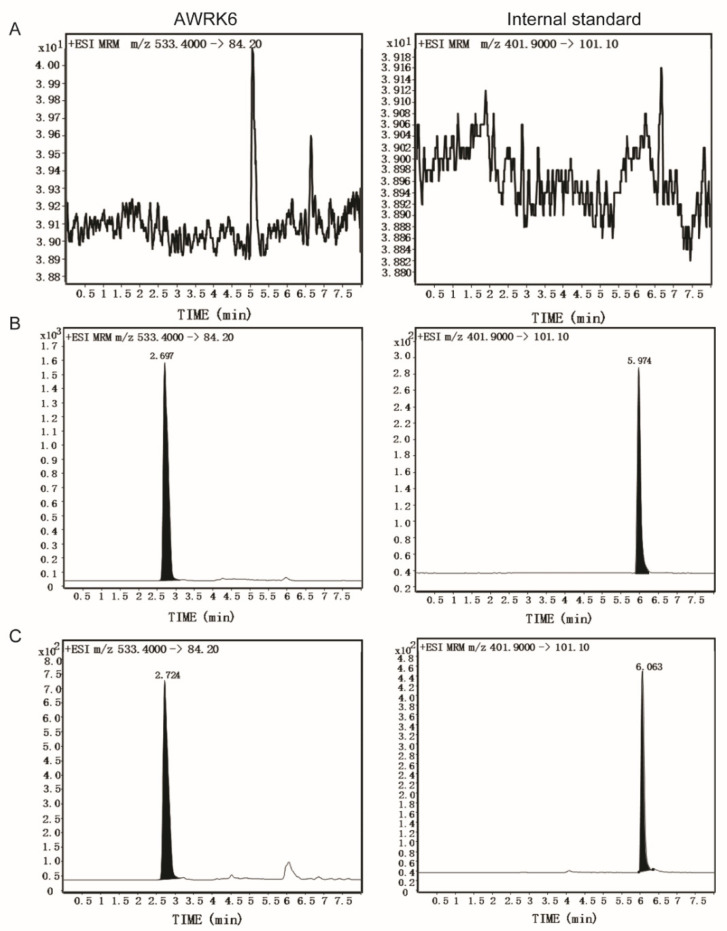
Chromatograms of blank plasma (**A**), spiked blank plasma (**B**), and a sample obtained at 30 min after intraperitoneal administration with AWRK6 at 4.7 mg/kg bodyweight (**C**).

**Figure 3 molecules-27-00092-f003:**
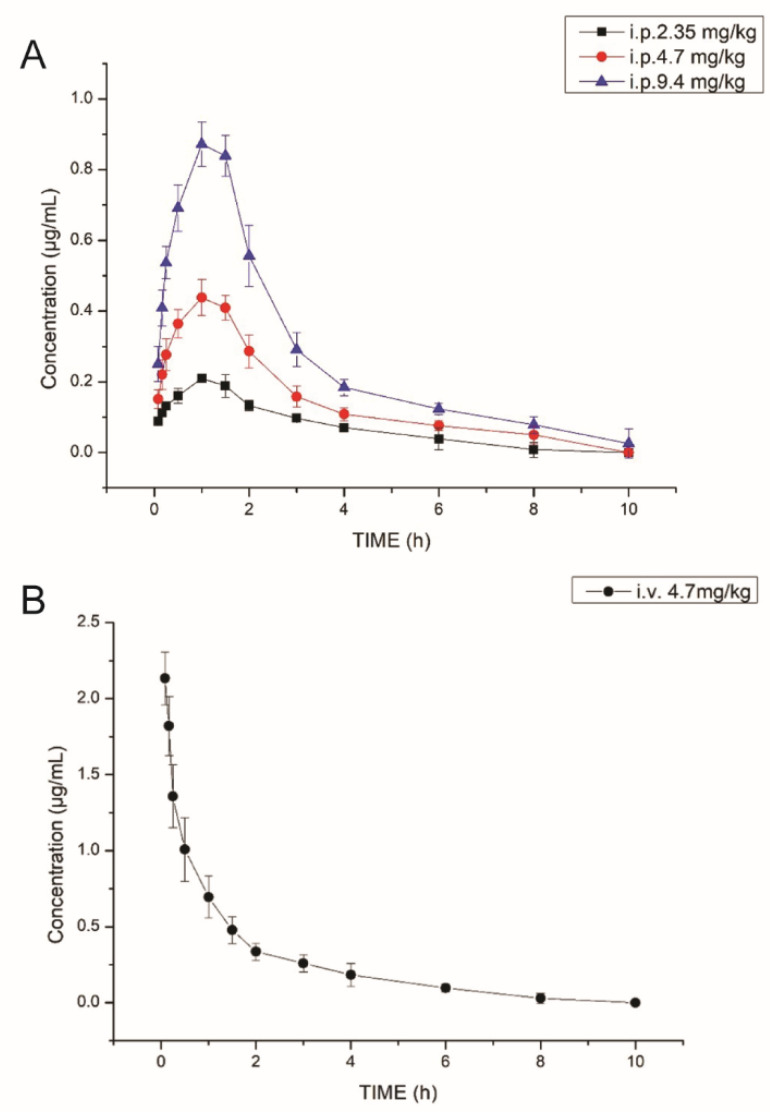
Plasma concentration time curves of intraperitoneal (**A**) and intravenous (**B**) administration. *n* = 6. The i.p. and i.v. indicate intraperitoneal injection and intravenous injection, respectively.

**Table 1 molecules-27-00092-t001:** Method validation for precision, accuracy, recovery, and matrix effect.

Concentration (μg/mL)	AWRK6 0.100	AWRK6 0.500	AWRK6 5.000	IS 1.000
Intra-day precision (% RSD)	2.726	3.132	1.245	-
Inter-day precision (% RSD)	2.970	3.490	3.185	-
Accuracy (% Normal)	87.000	90.600	92.060	-
Recovery (%) ^1^	82.178 ± 1.587	86.368 ± 2.189	84.600 ± 1.729	82.805 ± 1.821
Matrix effect (%) ^1^	−0.477 ± 2.268	+1.597 ± 2.943	+4.772 ± 4.661	−2.213 ± 0.725

^1^ Mean ± SD, *n* = 6.

**Table 2 molecules-27-00092-t002:** Stability of AWRK6 QC samples.

Concentration (μg/mL)	0.100	0.500	5.000
Room temperature for 6 h	0.089 ± 0.007 ^1^, −6.315% ^2^	0.462 ± 0.021, −4.742%	4.915 ± 0.027, −0.466%
Three freeze–thaw cycles	0.091 ± 0.008, −4.210%	0.477 ± 0.016, −1.649%	4.912 ± 0.049, −0.409%
Storage at 4 °C for 12 h	0.084 ± 0.013, −11.579%	0.442 ± 0.015, −8.866%	4.881 ± 0.301, −1.040%
Storage at −20 °C for 7 days	0.062 ± 0.018, −34.737%	0.387 ± 0.047, −20.206%	4.153 ± 0.442, −16.940%

^1^ Mean ± SD, *n* = 3. ^2^ Relative error (%).

**Table 3 molecules-27-00092-t003:** Pharmacokinetic parameters of AWRK6.

Dose (mg/kg)	2.350 (i.p. ^2^)	4.700 (i.p. ^2^)	9.400 (i.p. ^2^)	4.700 (i.v. ^2^)
AUC_(0–t)_ (mg/L/h)	0.634 ± 0.096 ^1^	1.365 ± 0.176	2.631 ± 0.236	2.598 ± 0.277
AUC_(0–∞)_ (mg/L/h)	0.851 ± 0.221	1.588 ± 0.275	2.914 ± 0.356	2.802 ± 0.333
MRT_(0–t)_ (h)	2.256 ± 0.488	2.491 ± 0.227	2.525 ± 0.292	1.733 ± 0.261
VRT_(0–t)_ (h^2^)	2.549 ± 1.480	3.859 ± 0.904	4.366 ± 1.439	-
t_1/2z_ (h)	2.946 ± 2.048	2.941 ± 1.399	2.781 ± 1.021	1.983 ± 0.583
T_max_ (h)	1.167 ± 0.258	1.250 ± 0.274	1.210 ± 0.217	-
CLz/F (L/h/kg)	2.879 ± 0.541	3.036 ± 0.529	3.263 ± 0.365	1.699 ± 0.219
Vz/F (L/kg)	10.95 ± 3.965	12.333 ± 4.853	12.738 ± 3.37	4.884 ± 1.736
C_max_ (mg/L)	0.217 ± 0.010	0.454 ± 0.048	0.888 ± 0.046	2.134 ± 0.174
F (%)	48.807	52.540	50.635	-

^1^ Mean ± SD, *n* = 6. ^2^ The i.p. and i.v. indicate intraperitoneal injection and intravenous injection, respectively.

## Data Availability

Not applicable.

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
