# Peer review of "Determination of the Peptide AWRK6 in Rat Plasma by Liquid Chromatography-Tandem Mass Spectrometry (LC-MS/MS) and Its Application to Pharmacokinetics"

_molecules, 2021, doi:10.3390/molecules27010092_

Round 1

Reviewer 1 Report

Manuscript entitled "Determination of the Peptide AWRK6 in Rat Plasma by LC- MS/MS and Its Application to Pharmacokinetics" by authors Lili Jin et al, describes about the PK profiling for Peptide AWRK6 in rats. This study was designed and explained well.

However, authors needs to include the rationale behind selection of these dose range for intraperitoneal and intravenous dosing. 

when it comes to matrix effect authors may explain how much percentage of ion enhancement or ion suppression effect like +8% or -5%. 

Discussion part: Authors could have included Cmax comparison with in vitro efficacy potency and plasma protein binding (if available)

Methodology: Was the intravenous doing administered via cannula or directly on the vein?. Please mention. 

Overall, this manuscript is written well to explain the details

Author Response

Dear Reviewer,

Thank you for your careful checking and professional comments. The manuscript has been revised according to those helpful suggestions. The changes were marked using built-in Track Changes in Microsoft Word and the main modifications are as following:

Reviewer 1

Manuscript entitled "Determination of the Peptide AWRK6 in Rat Plasma by LC- MS/MS and Its Application to Pharmacokinetics" by authors Lili Jin et al, describes about the PK profiling for Peptide AWRK6 in rats. This study was designed and explained well.

However, authors needs to include the rationale behind selection of these dose range for intraperitoneal and intravenous dosing.

Response:

Thank you for your suggestions. As you pointed out, the basis for the selection of the dose range used in pharmacokinetic studies should be clearly stated, especially for a novel drug candidate. Previously, in our study of AWRK6 alleviating LPS induced liver injury, the mice were injected intraperitoneally at a dose of 100 nmol/kg bodyweight (20 g per mouse). Calculated using the body surface area method, the corresponding administered dose for rats was 4.7 mg/kg bodyweight, which was used as the medium dose for intraperitoneal injection. And 9.4, 2.35 mg/kg body weight were used as the high and low doses, respectively. The intravenous dose was the same as the medium dose for intraperitoneal injection. The discussion section has been improved accordingly.

when it comes to matrix effect authors may explain how much percentage of ion enhancement or ion suppression effect like +8% or -5%.

Response:

Thank you for your suggestion. In Section 2.2 of the revised version, “… the matrix effects (98–105%) …” has been changed to “… the matrix effects (- 2.213% – + 4.772%) …”, and Table 1 has been modified.

Discussion part: Authors could have included Cmax comparison with in vitro efficacy potency and plasma protein binding (if available)

Response:

The in vitro efficacy potency and plasma protein binding were not studied. Your comments are helpful to our further study on the peptide.

Methodology: Was the intravenous doing administered via cannula or directly on the vein?. Please mention.

Response:

The intravenous doing was administered via tail vein of the rats. The Section 4.5 has been improved following your suggestion.

Overall, this manuscript is written well to explain the details

Thank you again for checking and providing those professional comments, which improved the manuscript and provided great help for our further research.

Kind regards,

Dianbao Zhang, PhD

China Medical University

Reviewer 2 Report

Reviewer report:

Line 49: LC-MS/MS; even it is well known, abbreviations should be explained upon their first appearance in the text.

Lines 177-179: references upon the synthesis of the peptide are missing.

Lines 190-191: An up-to-date  instrument is combined with an old-fashioned column (250x4.6); why not using a shorter and more efficient column in terms of chromatographic performance?

Results and discussion should be merged into a single section. The article is written in a rather old-fashioned way.

Tables 1 and 2: please correct the significant digits (decimal points)

Author Response

Dear Reviewer,

Thank you for your careful checking and professional comments. The manuscript has been revised according to those helpful suggestions. The changes were marked using built-in Track Changes in Microsoft Word and the main modifications are as following:

Reviewer 2

Line 49: LC-MS/MS; even it is well known, abbreviations should be explained upon their first appearance in the text.

Response:

Following your suggestion, the full name liquid chromatography-tandem mass spectrometry has been added for LC-MS/MS upon its first appearance in the revised version of the manuscript.

Lines 177-179: references upon the synthesis of the peptide are missing.

Response:

Thank you for your suggestion. One of our previous study has been added as the reference upon the synthesis of the peptide.

  1. Wang Q, Jin L, Wang H, Tai S, Liu H, Zhang D. AWRK6, A Synthetic Cationic Peptide Derived from Antimicrobial Peptide Dybowskin-2CDYa, Inhibits Lipopolysaccharide-Induced Inflammatory Response. Int. J. Mol. Sci. 19(2), (2018).

Lines 190-191: An up-to-date  instrument is combined with an old-fashioned column (250x4.6); why not using a shorter and more efficient column in terms of chromatographic performance?

Response:

Thank you for your professional comments. In earlier studies developing analytical methods, we used smaller Agilent and Waters columns such as 150 × 5 mm. However, the retention of the tested peptide was suboptimal, after which the column 250 × 4.6 mm was used for this study. We are beginners in the field of the development of methods for the peptide detection. In retrospect, perhaps other factors contributed to the failure of smaller columns. Thank you very much for your enlightening suggestions, which are very helpful for future assay optimization and further studies.

Results and discussion should be merged into a single section. The article is written in a rather old-fashioned way.

Response:

Thank you for your professional advice. This manuscript was formatted following the journal’s submission template, and we could consult with the editors about the style of the manuscript to suit the subject of this study.

Tables 1 and 2: please correct the significant digits (decimal points)

Response:

Thank you for your comments. The Tables content has been corrected following your suggestion.

Thank you again for checking and providing those professional comments, which improved the manuscript and provided great help for our further research.

Kind regards,

Dianbao Zhang, PhD

China Medical University

Reviewer 3 Report

The paper describes an LC-MS/MS method to study the pharmacokinetics profiles of a new synthesized peptide, denoted by AWRK6.  The novelty of this study assures the condition of being considered for publication, but the entire content is poorly organized and presented.  The paper needs a major improvement by a careful editing and presentation, since in many situations the phrases and notions are not correctly described.  A new version should include the following recommendations:

1)  The common order of the sections should be Introduction, Experimental, Results and Discussion, Conclusions and Refs.

2)  The part of Discussion section (lines 112-139) is not related to the subject of this study, and should be eliminated, or very shortly discussed in Introduction section.  

3)  The section dedicated to the sample preparation, which is very important for such a study, should describe clearly the principle of this procedure (is it plasma precipitation or liquid-liquid extraction ??, or a combination of these two possibilities, for example);

4)  In section 2.2. x and y should be defined;

5)  Matrix effects should be detailed by presenting the nature of (line 220); if they are spiked plasma samples, this is not correct.

6)  Standard solutions for calibration are obtained from water standards and blank plasma, in the ratio 50:100;  this ratio modify completely the matrix sample, diluting this to half;  therefore this aspect should be considered as very debatable.

7)  <… ACN presented shorter retention time and better peak shape> (line 142) should be reformulated to understand that the shorter retention time and better peak shape refer to the analyte and the internal standard.

8)  The composition of mobile phase (lines 191-193) includes (!!!, perhaps solutions).  The information at this page refers to the components of the mobile phase.   

9)  <… acceptable error limits> from Abstract should be replaced by confidence interval or alike, but this part is not necessary to be included in Abstract as not being relevant there;  more relevant could be information in Abstract about the chromatographic method and sample preparation procedure.  Also, from Abstract the second phrase is better to be eliminated.

10)  Data about precision of measuring IS signal (peak) should be given, such as in Table 1.

11)  The choice of internal standard should be discussed in section Results and Discussion. Short presentation from line 196 should be removed, as not being there its place.

12)  Besides LLOQ, the paper should include LOD and LOQ and how they were evaluated.

Author Response

Dear Reviewer,

Thank you for your careful checking and professional comments. The manuscript has been revised according to those helpful suggestions. The changes were marked using built-in Track Changes in Microsoft Word and the main modifications are as following:

Reviewer 3

The paper describes an LC-MS/MS method to study the pharmacokinetics profiles of a new synthesized peptide, denoted by AWRK6.  The novelty of this study assures the condition of being considered for publication, but the entire content is poorly organized and presented.  The paper needs a major improvement by a careful editing and presentation, since in many situations the phrases and notions are not correctly described.  A new version should include the following recommendations:

1)  The common order of the sections should be Introduction, Experimental, Results and Discussion, Conclusions and Refs.

Response:

Thank you for your checking of our manuscript. The manuscript was formatted following the journal’s submission template, and we could consult with the editors about the style of the manuscript to suit the subject of this study.

2)  The part of Discussion section (lines 112-139) is not related to the subject of this study, and should be eliminated, or very shortly discussed in Introduction section. 

Response:

Thank you for your suggestion. This paragraph (lines 112-139) has been eliminated in the revised version of the manuscript.

3)  The section dedicated to the sample preparation, which is very important for such a study, should describe clearly the principle of this procedure (is it plasma precipitation or liquid-liquid extraction ??, or a combination of these two possibilities, for example);

Response:

Thank you for your comments. Early in this study, the extraction of AWRK6 from plasma was attempted using liquid-liquid extraction with ethyl acetate and chloroform, a solid-phase extraction with SPE column, and the recoveries were extremely low. Protein precipitation was later carried out using acetonitrile and the recovery was also low. Later, methanol was used for precipitation and the recovery of AWRK6 was greatly improved. Then dichloromethane was added, and the recoveries reached 82.178% to 86.368%. Thus, a combination of plasma precipitation and liquid-liquid extraction was applied for the extraction of AWRK6 in samples. Section 4.3 has been improved accordingly following your suggestion.

4)  In section 2.2. x and y should be defined;

Response:

Thank you for your careful checking of the manuscript. Section 2.2 has been improved following your suggestion. It has been changed to “…the equation for the curve was y = 0.090541x + 3.311952 (r = 0.9957, x indicates the concentration of AWRK6 (μg/mL) and y indicates the ratio of the peak area of AWRK6 to IS) …”.

5)  Matrix effects should be detailed by presenting the nature of (line 220); if they are spiked plasma samples, this is not correct.

Response:

Thank you for your professional comment. In the revised manuscript, Section 4.4 has been improved following your suggestion. It has been changed to “Matrix effect was evaluated by comparing the respective peak areas of the QC samples with standard solutions (AWRK6 and IS in water respectively) at the same concentrations (repeated 6 times).”

6)  Standard solutions for calibration are obtained from water standards and blank plasma, in the ratio 50:100;  this ratio modify completely the matrix sample, diluting this to half;  therefore this aspect should be considered as very debatable.

Response:

Thank you for your comments. Consistent with the preparation of standard solution for calibration, the same method was used for QC samples and real rat plasma samples. Moreover, examination of matrix effects in the methodological validation revealed insignificant ion enhancement or suppression. Thank you for your critical comments, which will be very helpful for our future method optimization and further studies.

7)  <… ACN presented shorter retention time and better peak shape> (line 142) should be reformulated to understand that the shorter retention time and better peak shape refer to the analyte and the internal standard.

Response:

Thank you very much for your checking. “… ACN presented shorter retention time and better peak shape …” has been corrected to “… AWRK6 and IS presented shorter retention time and better peak shape in ACN …”.

8)  The composition of mobile phase (lines 191-193) includes (!!!, perhaps solutions).  The information at this page refers to the components of the mobile phase.  

Response:

Many thanks for your pointing out this error. The “solute” has been corrected to “solutions” in the revised manuscript.

9)  <… acceptable error limits> from Abstract should be replaced by confidence interval or alike, but this part is not necessary to be included in Abstract as not being relevant there;  more relevant could be information in Abstract about the chromatographic method and sample preparation procedure.  Also, from Abstract the second phrase is better to be eliminated.

Response:

Thank you for your kindly suggestions. The Abstract has been improved accordingly. The second phrase has been eliminated in the revised version, and it has been changed to “Here, a quantitative determination method for AWRK6 analysis in rat plasma by using liquid chromatography-tandem mass spectrometry (LC-MS/MS) was established and validated follow-ing U.S. FDA guidelines.” Moreover, the sentence “A combination of plasma precipitation and liquid-liquid extraction was applied for the extraction.” has been added to the Abstract.

10)  Data about precision of measuring IS signal (peak) should be given, such as in Table 1.

Response:

In this study, it was found that the in vitro stability of IS was ideal, and the chromatographic peak areas of IS from different samples were not significantly changed. Thus, the precision of IS was not assessed. Thank you for your suggestion, the precision of IS will be further examined in our later study to obtain the quantification of peptide more accurately.

11)  The choice of internal standard should be discussed in section Results and Discussion. Short presentation from line 196 should be removed, as not being there its place.

Response:

Thank you for your suggestion. As discussed in Discussion, An IS with similar physicochemical properties of the determined substance was desirable. The isotope-labeled analogue of AWRK6 was not available yet. Vancomycin, PMB, berberine, and an AWRK6 derivative (SKVWKHWKKFWHKAHKLH-NH2) were examined, and PMB was chosen as an IS for its superior recovery, matrix effect and chromatographic behavior. As you pointed out, the manuscript structure might be updated as needed.

12)  Besides LLOQ, the paper should include LOD and LOQ and how they were evaluated.

Response:

Thank you for your professional comments. The limits of detection (LOD) and limits of quantitation (LOQ) were estimated as concentrations of AWRK6 which generated S/N value of 3 or 10, and they were verified by continuous evaluating the S/N value of 3 LOD samples or 6 LOQ samples. The LOD and LOQ were 0.050 and 0.025 μg/mL, respectively. Section 2.2 and 4.4 have been improved accordingly.

Thank you again for checking and providing those professional comments, which improved the manuscript and provided great help for our further research.

Kind regards,

Dianbao Zhang, PhD

China Medical University

Round 2

Reviewer 2 Report

The revised version is suitable for acceptance.

Author Response

Dear Reviewer,

Many thanks for your kindly help to improve our manuscript. Your professional comments are very helpful for our follow-up research. We are beginners in pharmacokinetics, and the research will continue to deepen.

Dianbao Zhang

China Medical University

Reviewer 3 Report

The question 12 remained unsolved, and the answer is contradictory.  The parameters LOD and LOQ are defined in lines 199-200, but their values are calculated (line 73) in a total contradiction with these definitions.  Also, the value of LLOQ appears identical with LOD, which means it was incorrectly evaluated.  These aspects should be clarified and correctly discussed in a new version.

Author Response

Dear Reviewer,

Many thanks for your kindly help to improve our manuscript. The manuscript has been revised following your comments.

The question 12 remained unsolved, and the answer is contradictory.  The parameters LOD and LOQ are defined in lines 199-200, but their values are calculated (line 73) in a total contradiction with these definitions.  Also, the value of LLOQ appears identical with LOD, which means it was incorrectly evaluated.  These aspects should be clarified and correctly discussed in a new version.

Response:

Thank you for pointing out the error. The position of the two numbers was reversed. It has been corrected to “The limits of detection (LOD) and limits of quantitation (LOQ) were 0.025 and 0.050 μg/mL, respectively” in the revised version. LLOQ was equivalent to LOQ here.

Kind regards,

Dianbao Zhang

China Medical University